# Ecosystem Birth near Melting Glaciers: A Review on the Pioneer Role of Ground-Dwelling Arthropods

**DOI:** 10.3390/insects11090644

**Published:** 2020-09-19

**Authors:** Sigmund Hågvar, Mauro Gobbi, Rüdiger Kaufmann, María Ingimarsdóttir, Marco Caccianiga, Barbara Valle, Paolo Pantini, Pietro Paolo Fanciulli, Amber Vater

**Affiliations:** 1Faculty of Environmental Sciences and Natural Resource Management, Norwegian University of Life Sciences, 1433 Aas, Norway; 2MUSE-Museo delle Scienze, Corso del Lavoro e della Scienza, 3, I-38122 Trento, Italy; Mauro.Gobbi@muse.it; 3Department of Ecology, University Innsbruck, Technikerstrasse 25, A-6020 Innsbruck, Austria; Ruediger.Kaufmann@uibk.ac.at; 4Department of Biology, Lund University, Sölvegatan 37, S-223 62 Lund, Sweden; maria.ingimarsdottir@biol.lu.se; 5Department of Biosciences, Università degli Studi di Milano, Via Giovanni Celoria 26, 20133 Milano, Italy; marco.caccianiga@unimi.it (M.C.); barbara.valle@unimi.it (B.V.); 6Museo Civico di Scienze Naturali “E. Caffi” di Bergamo, Piazza Cittadella 10, I-24129 Bergamo, Italy; ppantini@comune.bg.it; 7Department of Life Sciences, University of Siena, Via Aldo Moro 2, 53100 Siena, Italy; paolo.fanciulli@unisi.it; 8Science and Technology Facilities Council, Polaris House, Swindon SN2 1FL, UK; amber_vater@hotmail.com

**Keywords:** arthropods, Collembola, food web, foreland, glacier, pioneers, succession

## Abstract

**Simple Summary:**

Due to climate change, glaciers are retreating. On newly deglaciated ground, ecosystems gradually evolve through the process of primary succession. This gives scientists a unique opportunity to study how a new ecosystem is born. During the first few years, before plants establish, the barren ground of sand, silt, and stones close to the ice edge is conquered by a rich variety of insects, spiders, and other small animals. Many of these are predators and their prey are either transported by air or produced in situ. The real pioneers are, however, wingless springtails that graze on biofilm containing algae or cyanobacteria. Studies of two pioneer food webs showed differences in structure and function. In the one case, beetles, spiders, and harvestmen exhibit preferences for locally produced springtails, while predators in the other example relied mainly upon midges hatching from young ponds. Pioneer communities contain a mixture of generalists and specialists. Species composition vary under different climatic and geographical conditions, depending on the available candidate species in the surrounding areas. This study illustrates flexibility in the early phase of primary succession. Certain cold loving beetles, spiders, and springtails may become extinct if glaciers disappear completely.

**Abstract:**

As glaciers retreat, their forelands represent “natural laboratories” for the study of primary succession. This review describes how certain arthropods conquer pristine ground and develop food webs before the establishment of vascular plants. Based on soil samples, pitfall traps, fallout and sticky traps, gut content studies, and some unpublished data, we compare early arthropod succession on glacial forelands of northern Europe (Iceland, Norway including Svalbard, and Sweden) and of the Alps (Austria, Italy). While macroarthropod predators like ground beetles (Coleoptera: Carabidae), harvestmen (Arachnida: Opiliones), and spiders (Arachnida: Araneae) have usually been considered as pioneers, assumed to feed on airborne prey, this review explains a different pattern. Here, we highlight that springtails (Collembola), probably feeding on biofilm made up of algae or cyanobacteria, are super-pioneers, even at high altitudes and under arctic conditions. We also point out that macroarthropod predators can use locally available prey, such as springtails or non-biting midges (Diptera: Chironomidae). Pioneer arthropod communities vary under different biogeographical and climatic conditions. Two pioneer food webs, from northern Europe and the Alps, respectively, differed in structure and function. However, certain genera and orders were common to both. Generalists and specialists live together in a pioneer community. Cold-adapted specialists are threatened by glacier melting.

## 1. The Questions

Due to climate change, glaciers are retreating [1,2,3]. The freshly deglaciated areas left behind after glacier retreat represent unique “ecological laboratories” that can be used to study primary succession (Figure 1 and Figure 2). We observe how pioneer species arrive, how communities establish and how food webs develop. While the establishment and succession of plants has been well studied in glacial forelands [4,5,6,7,8,9,10], the colonization of animals has been investigated less. Here, we summarize and discuss recent European studies (Figure 3) on the role of arthropods during early succession. The number of years since deglaciation varies between studies but is always less than 50 and several studies focus on the very first few years. Figure 1 illustrates a large retreat during one summer [11]. These small animals reveal a great ability to conquer virgin ground and in certain ways challenge our ecological thinking.

We discuss how the demands of various pioneer species can be fulfilled within the harsh and unstable environmental conditions on newly deglaciated ground. How do local conditions in habitat structure, shelter, climate, and food availability influence colonization? We describe how species must pass through certain “ecological filters” to become pioneers and ask whether they are specialists or generalists. Furthermore, why are the taxonomic structures of pioneer arthropods communities within a restricted geographical area rather predictable? We also discuss why certain early species are gradually pushed-out of the community, challenging the principle of persistence after colonization [12].

A much-debated topic during the last two decades is the “predator first paradox”: How is it possible for predators to establish almost immediately, before the presence of vascular plants and herbivores? It has been hypothesized that early predators like carabid beetles and spiders feed on airborne prey, and organic material blown in from the surrounding area supports scavenging detritivores like Collembola [13]. Here we present observations that shed new light on the “predator first paradox”.

From community structure we go to discuss function. What do we know about early food webs? We compare two well-described pioneer food webs based on gut content studies—one from northern Europe (Norway) and one from the Alps (Austria). Both studies claimed to have “solved” the predator first paradox by documenting local production of prey, although different types of prey.

At the end, we will try to answer two main questions about early succession:A.How “flexible” are arthropod communities in the early phase of primary succession?B.Do pioneer arthropod communities under different conditions still have certain basic traits in common?

## 2. Conquering Newly Deglaciated Ground: A Passage through Ecological Filters

Early work introduced eight categories of colonizers, each characterized by their preference for food or substrate [14]. Here we present the colonization process of pioneer species in a slightly different light, considering a wider range of factors. Each pioneer species has to overcome certain obstacles, or “ecological filters”, before it is able to establish its presence. Firstly, it must travel, which demands a sufficient dispersal ability. Secondly, it must find an acceptable habitat with tolerable climate and shelter to avoid hostile conditions. Thirdly, it must be able to find food and finally, a true pioneer species must also be able to reproduce. We now explore these filters further.

### 2.1. Ability to Disperse

Existing literature contains several observations on fallout of winged insects on glaciers and snowfields [13]. In alpine snow fields and glaciers of southern Norway and the Alps, as well as on Icelandic nunataks, even typical lowland-insects have frequently been recorded, lifted upwards by warm air currents (Hågvar, Kaufmann, and Gobbi, pers. obs.) [15,16,17,18].

The isolation of nunataks makes them especially interesting in studying the dispersal of non-winged arthropods. In Iceland, the microarthropod fauna of isolated nunataks was compared with that of non-isolated deglaciated areas [19]. They found that isolation of a few kilometers did not significantly affect the colonization of Collembola and oribatid mites, and it was assumed that these tiny animals were transported by wind.

We know therefore that dispersal occurs, but it is often difficult to quantify this. Evidence from Austrian fallout traps in Rotmoos valley produced Collembola catches and plant material; however, a reliable quantification was not possible [18]. In Norway, 30 fallout traps (diam. 6.5 cm) were operated during a four-week period on a five year old moraine [20]; 108 microarthropods: 26 Acari and 82 Collembola were caught. The total fallout in this period was calculated to 1080 microarthropods per m^2^. Other arthropods in the traps included Diptera, especially chironomid midges, but also some wingless aphids. Interestingly, plant organic material in fallout traps included fragments and diaspores of mosses, often in considerable quantities, and of which several fragments had the potential to start new colonies.

Sticky traps in the same site showed that both arthropods and mosses were transported close to the ground, below about 40 cm height [20]. On a glacier foreland on Svalbard, nearly all flying insects which colonized pioneer ground were trapped below 25 cm height [21]. It was explained by a favorable microclimate with little wind near the ground, and that the “aerial transport” in this case was mainly due to active flight. On Iceland, sticky traps placed at 30 cm height on a medial moraine (debris cover that stretches from nunatak down-glacier to lowland) collected many Diptera compared to traps placed on white ice, indicating that active flight occurs over ice-free ground [16]. Active flight of winged insects on glacier forelands clearly occurs mainly near the ground but it is easy to imagine that episodic storms may carry both winged and wingless organisms high up and long distances.

Linyphiidae spiders are known to be efficient aerial dispersers, having the ability of long distance “aerial ballooning”, hanging on silk threads. This has been well demonstrated in glacial forelands in Svalbard [13,21,22].

Certain Collembola are probably able to keep pace with the receding glacier front by jumping and walking. In Norway, the cold-adapted Collembola *Agrenia bidenticulata* was observed to make jumps on the surface of cold water, and after a glacier retreat of 30 m one year it was actively present 4 m from the ice edge [20]. On partly vegetated ground of 34–38 years age, pitfall catches revealed that Collembola were not confined to vegetated plots but showed a high surface activity on bare ground [23]. This is also true on young sites of Italian glacier forelands, even if the bare ground is coarse, highly irradiated and percolating and the dispersion seems not to be easy [10,24]. High surface activity of Collembola on newly deglaciated ground was likewise documented in Austria [25]. Furthermore, on Icelandic nunataks, pitfall catches documented that Collembola were active on bare ground, even on land younger than 10 years old [19].

The large and fast-moving harvestman *Mitopus morio* can easily trace the receding glacier on foot [26]. This may be true for all mobile macroarthropod colonizers. However, recently emerged Icelandic nunataks seem to be too isolated by snow and ice to be reached by *M. morio* [15].

### 2.2. Ability to Find Acceptable Habitat Qualities

In order to survive, arrivals must be able to avoid or overcome extreme microclimates including temperature and moisture extremes. Arrivals may also face surface flooding by meltwater, or unconsolidated soils and soil movement. Finding shelter to avoid these extremes, including use of sub-surface habitats between stones and gravel, will be important. An inability to avoid extremes, will mean non-survival. Bare rock surfaces are unfavorable. However, rock crevasses may be safe sites for colonization of arthropods.

Figure 4 visualizes high spatial heterogeneity on an eight-year-old moraine at Hardangerjøkulen glacier in Norway that creates a variety of microenvironments and habitats [27]. The morphologically varied terrain contained various surface structures, from fine-grained silt, sand, and gravel to larger stones and boulders, sometimes in a complex matrix. Fast-moving predators, like ground beetles and spiders hunt on such surfaces. Young ponds (some less than one m^2^ in size) contained sediments rich in larvae of Chironomidae and Tipulidae. In the porous limestone bedrock areas of the Alps, the silt and clay proportions of pioneer ground are usually scarce compared to gravel and sand. In front of the Trobio glacier, soils younger than 30 years contained about 3% of silt and clay [10]. Ponds are less frequent because of the well-drained soil, however Chironomidae are able to colonize glacial streams and supraglacial lakes [28]. Vertical labyrinths created between stones and gravel and below surface stones, may offer shelter during day for nocturnal carabids and habitats for their larvae [29]. In Rotmoos valley, Austria, carabid larvae have been caught in specially designed traps, positioned at 0.5 m depth below the surface (Kaufmann, unpublished).

Microclimate and structural diversity are two closely connected parameters. For example, during sunny days, open ground, including stones, can be considerably heated due to the lack of shadow-producing vegetation. Heat can be slowly released afterwards, improving hunting conditions for predators and even some nocturnal species. Shallow ponds may also be considerably heated by sun. On windy days, stones create shelters, both horizontally and vertically. Figure 5 illustrates how wind-dispersed diaspores of pioneer mosses can aggregate and thrive alongside a large stone [30]. The effect of local microclimate for early succession has been well documented in the Austrian Rotmoos foreland [9,31]. Even small temperature changes by climate warming may profoundly speed up the initial colonization [32].

On the barren moraine surfaces the situation is extreme, and in Rotmoos, daily amplitudes from −10 to + 60 °C locally have been measured in June. This heat stress, occurring together with desiccation pressure, requires good strategies for the colonizers to avoid the extremes, and to optimize their foraging. Winter under the snow cover is favorable, not only for the microbiota but also for certain carabids that are active beneath snow. Snow-free periods with the ground freezing at −30 °C will kill carabid larvae, consequently low adult numbers are present the next summer. Once a plant cover has developed, the situation becomes much less extreme.

Response to local environmental conditions can be demonstrated on the small scale in pitfall grids. In Rotmoos valley, the occurrence patterns of various carabid species remained stable over years. A non-continuous occurrence of *Nebria jockischii* illustrates the point: local conditions are decisive, not only site age (Figure 6, unpublished).

Factors like these can create local successional pathways, finally leading to specific occurrence patterns for carabids and other arthropods on the landscape scale [33]. In Norway, dry and wet patches have been shown to be colonized by different arthropods and to develop different succession patterns [34].

### 2.3. Ability to Find Food

To a certain degree, airborne arthropods will act as food for early predators and contribute to energy flow in early food webs, but prey like Collembola and chironomid midges can also be produced locally [17,27]. Chlorophyll may appear surprisingly early, for instance as biofilm with diatom algae or cyanobacteria, or as tiny pioneer mosses [11,20]. Furthermore, bacteria and fungi may be food sources on pioneer ground. This topic will be further explored in a later section using two case studies to compare pioneer food webs. The case study from Hardangerjøkulen glacier in Norway documented a variety of food sources on 3-6 years old ground, and different pioneer arthropods selected among these options (Table 1). Studies on the Austrian Alps have described a high intraguild predation as well as a heavy predation on Collembola [17,35].

### 2.4. Ability to Reproduce

Both adults and young stages of Acari and Collembola were found on pioneer ground near Hardangerjøkulen in Norway, indicating reproduction. Young stages may, however, have been transported by wind. On a six year old Norwegian moraine, a larva of the beetle *Simplocaria metallica* (Byrrhidae) was recorded in a small patch of pioneer moss (Figure 7, [30]). Since carabid larvae may have a cryptic life between gravel and stones, we assume that *Nebria nivalis* and *Bembidion hastii* reproduced close to the glacier. Adults of both species were trapped on three year old ground, the later in large numbers [36]. From the Rotmoos glacier foreland in Austria, larvae, and juvenile stages within the first years of deglaciation have been identified for carabid beetles of the genera *Nebria* and *Oreonebria*, the spiders *Pardosa nigra* and *Erigone tirolensis*, as well as the harvestman *Mitopus glacialis* [31]. *Nebria* and *Oreonebria* seem to have a two year larval development [33] and long-lived adults (at least three years according to mark-recapture results (Kaufmann, unpublished)). Prolonged life-cycles are typical for high altitude species and/or populations [37], but their role in glacier forelands is unclear. Juvenile stages of species belonging to the genera *Nebria* and *Oreonebria* were also found in early successional stages and on supraglacial debris of Forni glacier in the Central Italian Alps [38], Amola glacier in the Central Eastern Italian Alps [39], and the Agola glacier of the Western-Dolomites [40]. From the Alps, the carabid *Pterostichus jurinei* is an example of a predator that by some reason is unable to reproduce on pioneer ground. It was found occasionally in single spots in the pioneer zone, but never established populations there (Kaufmann, unpublished) [41,42]. In later stages it was common [43].

On nunataks in Iceland, *Amara quenseli* larvae were caught in pitfall traps on land less than three years old and on 10–35 years old land. Further, Diptera larvae (Cyclorrapha) were occasionally caught in non-baited pitfall traps on a debris-covered glacier and on land younger than 10 years (Ingimarsdóttir, unpublished).

Along the Amola glacier foreland (Italian Alps), there is evidence of a trend in population sex-ratio of *Nebria germari* in relation to time since deglaciation, and thus distance from the glacier [29]. The sex-ratio of *Nebria germari* was female-biased on the debris-covered glacier and in front of the glacier snout. Probably, the initial colonization from the early successional stages (source habitat) to the glacier surface was favored by founder females (i.e., females arrived already fertilized and laid their eggs between the stones), which had a tendency to disperse. In Rotmoos, *Mitopus glacialis* seems to be parthenogenetic on the glacier foreland (Kaufmann, unpublished).

### 2.5. Different Filters for Different Successional Stages

Later successional stages will have other filters. An herbivore will, for instance, depend on the presence of its food plant, and a specialized predator or parasite must find its prey or host. Even if suitable food is present, the microclimate or habitat structure may be unacceptable. In an Icelandic study, dispersal ability and colonization success were studied in combination [19]. Microarthropods arriving more or less randomly on isolated nunataks (Figure 8) only survived on ground with a suitable age for each species, and a successional pattern was established similar to that seen in non-isolated forelands.

## 3. Which Arthropod Taxa are Present on Pioneer Ground? A Comparison between Northern Europe and the Alps

The following lists of pioneer species on recently deglaciated ground (Table 2, Table 3, Table 4 and Table 5) are mainly based on pitfall trapping (Figure 9) but also on soil samples for microarthropods. Macroarthropod lists cover surface active beetles (Coleoptera), harvestmen (Opiliones), and spiders (Araneae, Figure 10). Flying insects are not included here. Sampling intensity, age of the ground, and taxonomic resolution varies but these lists still tell a lot about early arthropod communities in different European countries and sites. During the compilation, it became evident that both the micro- and macroarthropod fauna of the Alps differed from that of northern Europe. In addition to this latitudinal diversity, longitudinal differences in species composition have been demonstrated in the Alps [44]. In the following section, we treat northern Europe and the Alps separately. Within each area, we look for characteristic pioneer species and try to understand early succession. Afterwards, we will compare the two geographical areas in search of common traits.

### 3.1. Microarthropods: The Super-Pioneers

#### 3.1.1. Northern Europe

Microarthropods are probably the first animals to colonize deglaciated ground. A compilation of data from young forelands in Iceland, southern Norway and Svalbard illustrates their rapid presence (Table 2). Even a high Arctic foreland on Svalbard was inhabited by four species of Collembola, one Oribatida mite species, as well as Gamasida mites after two years. In southern Norway, eight Collembola species were recorded on 0–3 years old ground. One of them, the cold adapted *Agrenia bidenticulata*, followed the retreating ice edge closely and showed jumping activity on a melt water surface.

Among the long list of Icelandic pioneer Collembola, three were common to Hardangerjøkulen in southern Norway (*Desoria olivacea, Desoria tolya* and *Lepidocyrtus lignorum*), and two were common to Svalbard (*Folsomia quadrioculata* and *Isotoma anglicana*). Among Oribatida, *Tectocepheus velatus*, a well-known pioneer species, occurred in all three sites. Wind transport of this species was proved by sticky traps on a young moraine near Hardangerjøkulen in Norway [20]. On genus level, *Desoria, Lepidocyrtus, Folsomia*, and *Isotoma* are good North-European pioneer candidates among Collembola, and the very small *Liochthonius* species among Oribatida. Furthermore, Prostigmata and Gamasida mites were often observed in pioneer communities.

#### 3.1.2. The Alps

In the Austrian Alps, a number of Collembola species have been recorded in the Rotmoos foreland on ground younger than 40 years [47] and from younger ages down to 0–3 years [25]. Older Austrian data exist from Hintereis nearby [42]. From Italy, unpublished data on pioneer Collembola are available from four different forelands, among them the glacier Amola (Figure 11). Table 3 shows that several of the Collembola recorded in the Alps belong to other genera than in northern Europe. Some species were observed on the glacier surface, often on debris-covered glaciers. Furthermore, the species list reveals only few common species to Austria and Italy: *Heterosminthurus diffusus, Orchesella* cf. *alticola*, and *Isotomurus palliceps*. Even within the Italian Alps, some species change along a longitudinal gradient, suggesting local variation in species composition. Acari commonly occur on recently deglaciated terrains of the Italian Alps but have not yet been studied in-depth.

### 3.2. Macroarthropods: The Early Predators

#### 3.2.1. Northern Europe

Most early macroarthropods are predators. They are typically a mixture of harvestmen, carabid beetles, and various spiders, usually Lycosidae and Linyphiidae (Table 4). The high arctic site on Svalbard had only linyphiid spiders in addition to microarthropods.

Eight forelands younger than 20 years in Jostedalen and Jotunheimen in Norway had a rather predictable pioneer community [48]. Several of these species were common to a three year old site at Hardangervidda. A foreland near Veslejuvbreen glacier in the high alpine zone, however, contained only two beetles and two linyphiid spiders. Some predator species within the genera *Nebria, Bembidion, Mitopus, Erigone*, and *Collinsia* were common to Norway and Sweden. Icelandic forelands younger than ten years contained linyphiid spiders. However, a very old and partly vegetated nunatak contained three of the “classic” pioneers of Norway: *Mitopus morio*, *Amara quenseli*, and a *Pardosa* wolf spider [15].

#### 3.2.2. The Alps

Pioneer macroarthropods from the Alps are listed in Table 5. Many of the species are noted only from either Austria or Italy, and several are endemic and/or cold adapted. Pioneer macroarthropods are typically a mixture of harvestmen, carabid beetles and various spiders, usually Lycosidae and Linyphiidae. Centipedes (Chilopoda) were sometimes caught by pitfall traps on supraglacial debris and pioneer grounds of Italian Alps (*Lithobius lucifugus* on Sorapiss glaciers) but usually they are associated with older sites [31,51]. Lepidoptera larvae have been found abundantly on Rotmoos foreland after 34 years of deglaciation [47]. These data are consistent with the hypothesis that they are the initial humus formers [42].

Younger sites of Alpine glacier foreland and supraglacial debris host endemic cold-adapted species, for instance *Nebria germari* [39,40], *Oreonebria soror tresignore* [10], and *Mughiphanthes brunneri* [43].

### 3.3. Comparing Pioneer Communities in Northern Europe and the Alps

On genus level, young glacier forelands in the Alps contain a macroarthropod fauna that has much in common with forelands in Norway and Sweden (Table 4 and Table 5). Common genera for the two areas are *Nebria* and *Amara* among Carabidae beetles, *Simplocaria* among Byrrhidae beetles, *Mitopus* among Opiliones, and *Pardosa*, *Erigone, Meioneta*, and *Collinsia* among Aranea. Table 5 also illustrates how the genera *Nebria*, *Oreonebria*, *Amara*, and *Pardosa* occurred in different localities of the Alps. On species level, however, few are common to Northern Europe (e.g., Norway and Sweden) and the Alps: *Amara quenseli, Erigone tirolensis*, and *Meioneta nigripes.* Among pioneer Collembola, northern Europe and the Alps have several genera in common but only *Isotoma viridis* and *Micranurida pygmaea* on species level according to Table 2 and Table 3. In general, Isotomidae and Hypogastruridae were two characteristic pioneer families for northern Europe and the Alps. Surface active Symphypleona may also be early colonizers, present after only two years on Svalbard, after 1–3 years in Austria, and after three years in Norway. Certain Symphypleona were found on supraglacial debris in certain Italian sites (Table 3). Oribatida mites have been documented among pioneers in northern Europe, but information from the Alps are missing. Data from Rotmoos foreland in Austria indicate that Oribatida become abundant in the >30 year stages.

The faunal difference between northern Europe and the Alps can be related to Pleistocene glaciations. While the fauna in Norway, Sweden, Iceland, and Svalbard had to recolonize after the last glaciation, glacial and interglacial refuge areas located in the Alps during the Pleistocene drove speciation. A number of cold adapted arthropods and plant evolved in the Alps [54].

Moreover, Table 2, Table 3, Table 4 and Table 5 reveal great difference in both macro- and microarthropod fauna within the Alps, between the Austrian and Italian study sites. For example, some species belonging to the *Oreonebria* genus (e.g., *Oreonebria soror*), as well as several species of spiders, are steno-endemic of restricted geographic areas. It is also possible that Collembola underwent similar speciation trends during the Pleistocene; this is suggested by the variability that is observed along the southern Alps. The percentage of cryptic species of Collembola has been calculated as very high [55]. For example, in Rotmoos valley, two cryptic species of *Isotomurus alticolus* were recorded [25]. This may indicate ongoing speciation, and biodiversity should be analyzed on a molecular level.

We are led to the conclusion that only glacier forelands within a restricted geographical area have a rather predictable fauna of pioneer arthropods. Examples on such restricted areas are Norway/Sweden/Iceland, closely situated foreland valleys in Austria, and forelands in northern Italy belonging to the same biogeographical area. On the genus level, however, there are some common traits across Europe.

### 3.4. Effects of Altitude and Latitude on Pioneer Fauna

Up to about 1400 m altitude in Norway and Sweden, the pioneer fauna in glacier forelands is rather characteristic, containing both Opiliones (*Mitopus morio*), certain carabid beetles and various Lycosidae and Linyphiidae spiders. However, at the high alpine site Veslejuvbreen in Norway, at 1860 m altitude, the only macroarthropods recorded were two beetles and two Linyphiidae spiders (Table 4). Furthermore, succession is slower at high altitudes [12,49,56].

With increasing latitude, there was a gradual “thinning” of pioneer species. In Iceland, young forelands (less than ten years old) contained few macroarthropod species. However, on a large and old, partly vegetated nunatak, certain “classic” pioneer taxa were present on young soil: the harvestman *Mitopus morio*, the carabid beetle *Amara quenseli*, and Lycosidae and Linyphiidae spiders. In the high Arctic on Svalbard, Opiliones, beetles, and Lycosidae were absent, but certain Acari and “ballooning” Linyphiidae spiders still joined Collembola in colonizing during the first two years [14].

In the Alps, the investigated areas deglaciated less than 20 years ago (Table 3 and Table 5) were located at an average altitude of 2500 m, thus usually above the treeline. The Alps lack large plateau glaciers that provide the ice masses needed for a glacier tongue to reach far down into the valleys; however, there are a few big debris-covered glaciers that reach very low altitude (e.g., Miage glacier and Belvedere glacier that reaches the coniferous forest at 1800 m asl). Thus, the differences in species composition among the investigated glacier forelands can be explained by different biogeographic patterns and not only by habitat filtering [24]. On higher taxonomic level, the pioneer fauna in Alpine glacier forelands is rather similar to those of northern Europe, with Opiliones (*Mitopus* sp.), certain carabid beetles (*Nebria* spp.), springtails, and various Lycosidae and Linyphiidae spiders.

In summary, the harsher climate related to altitude or latitude gradually excludes most carabid beetles, Lycosidae, and Opiliones in Europe. Collembola, however, seem to thrive and reproduce on pioneer ground both under arctic and high alpine conditions.

## 4. From Structure to Function: A Comparison between Two Well-studied Pioneer Food Webs

Having described variations in the taxonomical structure of pioneer arthropod communities, this section focuses on function, i.e., food choice and food web architecture. Two thorough studies exist on pioneer food webs close to receding glaciers, one in Norway [57] and one in Austria [17]. These two forelands are comparable as they are both above the tree line, the actual terrain was deglaciated less than eight years ago, and they have a similar arthropod fauna on higher taxonomic level. We are interested in the following aspects:-If the two food webs are different: why are they different, and can the difference illustrate flexibility in pioneer arthropod communities?-Can these well-studied pioneer food webs shed new light on the “predator first paradox”?

### 4.1. Study Sites and Methods

The Norwegian study was performed close to the receding Hardangerjøkulen glacier in the southern, alpine part of the country [57]. Gut contents of three predators were studied in the microscope: the carabid beetles *Nebria nivalis* and *Bembidion hastii* and the harvestman *Mitopus morio*. Good knowledge of the anatomy of potential prey at the site allowed for identification of chitinous prey fragments in crop and gut. Early herbivory on pioneer mosses or biofilm was documented through gut content studies of certain Collembola, the Byrrhidae beetle *Simplocaria metallica*, and two omnivore carabid beetles, *Amara quenseli* and *A. alpina*. Species or genus of moss could often be identified from the cell structure in moss leaf fragments, even in Collembola guts [11].

The Austrian case was a DNA-based presence/absence study of prey items in gut contents of several predators [17]. The importance of a certain taxon, for instance Collembola, as prey was indicated by the percentage of predator guts that contained the actual taxon. Two early age stages (0–8 and 13–20 years old) were investigated in three glacier forelands in neighboring valleys in Tyrol, among them Rotmoos valley with a long research history (Figure 2 and Figure 12). Extensive DNA analyses of gut contents made it possible to construct food webs.

### 4.2. Main Similarities and Differences

The Austrian study found great similarities in the food webs of the three valleys, and also between the two age classes of the ground. The overall conclusion was as follows: the three food sources, Collembola, other predators, and flying insects, contributed with approximately one third each, to the food consumed by predators. The Norwegian study concluded the opposite, with few Collembola eaten, and few other predators on the menu. Instead, chironomid midges belonging to local decomposers, were the dominant prey (Table 6).

Certain conclusions were, however, common in the Austrian and Norwegian study:Typical macroarthropod predators in the pioneer community were Carabidae beetles, Opiliones and Lycosidae, and Linyphiidae among spiders.The main food for pioneer predators was considered to be produced locally and not transported to the pioneer ground by air.Collembola seemed to be among the very first animal colonizers of newly exposed ground.

### 4.3. Special Observations in the Austrian Case

All predators were flexible generalists. This led to rather complicated food webs. Still, the main web structure was surprisingly similar between sites and ages. Intraguild predation (predators eating other predators) was documented to be high, Collembola was often a favored prey, and several groups of airborne, winged insects were also eaten. The situation can be visualized as a hunting ground rich in predators, Collembola and landed, winged insects. Assuming to be resident, with reproducing populations on pioneer ground, Collembola were probably key organisms for rapid establishment of predators. Remarkable was a pronounced maximum of *Heterosminthurus diffusus* immediately at the glacier front (Table 7) [25]. Most of the Collembola analyzed had guts filled with various fungal material, mineral particles, and airborne pollen. A Symphypleona maximum at glacier edge was confirmed by [18]. In contrast to the Norwegian case, chironomids were not abundantly available. In all three valleys, there were only few and small spring fed brooks hosting insect larvae. The glacial stream is hostile for insects due to its silt load. Non-biting midges are the main colonizers of glacier-fed streams, with some species belonging almost exclusively to the genus *Diamesa* [28,58].

Despite being all generalists, the predators differed in their usage of prey items, and they responded positively to periodic high availability of specific prey, for instance a pulse of airborne aphids. There were some predator specific diet shifts from the glacier front to the following stage with the first pioneer plants, most notably that the consumption of collembolans decreased in one of the carabids and in both wolf spider species [17]. Over longer successional times, however, such diet shifts can become much more pronounced as shown by analyses of stable isotopes [59].

### 4.4. Special Observations in the Norwegian Case

The Norwegian study revealed a surprisingly early presence of chlorophyll. The very start of a community was terrestrial biofilm with photosynthesizing diatom algae. The cold-loving, very active Collembola species, *Agrenia bidenticulata*, grazed on the biofilm and was able to follow the retreating ice edge closely. The species was absent in later successional stages (Table 8). Tiny pioneer mosses were established after four years and was grazed upon by the large Collembola *Bourletiella hortensis*, the Byrrhidae beetle *Simplocaria metallica*, as well as the omnivore Carabidae beetles *Amara alpina* and *A. quenseli*.

The local production of chironomid midges in the Norwegian site had a surprising explanation. Chironomid larvae developed in the sediment of small, young ponds, where they ate and assimilated bioavailable, ancient carbon released by the glacier. The ancient carbon compounds were assumed to be long-transported aerosols resulting from incomplete combustion of fossil fuels [60]. Chironomid larvae, with ancient carbon embedded in their tissue, had a radiocarbon age of 3270 years. Adult chironomids with a radiocarbon age of 1400 years transported the ancient carbon to terrestrial predators, which achieved radiocarbon ages between 340 and 1100 years [27,57].

*Isotoma viridis* and *Lepidocyrtus lignorum* are two well-known generalists among Collembola. They colonized early and remained for a long time. Table 9 illustrates their ability to adjust their food choice according to what is available. On three year old ground, mineral particles dominated their gut content, and biofilm feeding is probably a good guess. On 30–40 years old ground, some fungal hyphae and spores were seen in guts, and on 63 years old ground, both species had become typical fungal feeders. Clearly, pioneer ground is open for both specialists and generalists among Collembola [11].

### 4.5. Discussion of the Two Studies

#### 4.5.1. Flexibility in Pioneer Food Webs

This comparison between the two forelands illustrates flexibility in food webs on newly deglaciated ground. Based on the same main groups of arthropods (Carabidae, Lycosidae, Linyphiidae, Opiliones, and Collembola), two different food webs were established, taking advantage of different local resources. In both sites, generalist predators adjusted their food according to what was available. While the Norwegian site was favored by local production of chironomid midges as prey, the Austrian case illustrated a mixed diet: resident Collembola and other predators, as well as airborne prey, including pulses of aphids.

Flexibility also characterized the Collembola fauna, both in succession pattern (Table 7 and Table 8) and in the food choice of single species (Table 9). The Norwegian study documented early chlorophyll by terrestrial diatom algae and tiny pioneer mosses, and pioneer Collembola fed on these sources. The food webs described from Austria did not include chlorophyll of any kind. However, diatom algae, pollen, and fungal hyphae were seen in collembolan guts [25]. A diverse fungal community can exist close to glaciers, more resembling cryoconite and glacier surface than typical soil communities [61]. Furthermore, unpublished studies of Collembola guts from the Italian Dolomites have indicated the presence of cyanobacteria. In summary, pioneer ground may contain a variety of food options for Collembola.

#### 4.5.2. The Predator first Paradox Resolved?

The paradox relies on two assumptions: That predatory macroarthropods are first, and that their food is transported into the pioneer ground through the air [13]. Studies from both Austria and Norway indicated that Collembola, and not predators, are the real pioneers. Moreover, both studies documented that an important part of the available prey can probably be produced locally—either as resident Collembola or as chironomid midges hatching from young ponds or rivers. Future studies in other forelands may well document cases where prey is mainly airborne, so we shall not exclude that the “paradox” may be explained in that way. Anyhow, for an external observer, the presence of predators on pioneer ground without visible plants or herbivores, will spontaneously appear as a biological paradox.

## 5. General Discussion

### 5.1. Driving Forces in Early Succession: A Geoecological View

In early succession theory, the concepts of facilitation, inhibition, and tolerance were introduced to illustrate how species favored or inhibited each other [5]. This view was purely biotic. Later, a more fruitful geoecological perspective was introduced, taking abiotic factors into consideration. One aspect is that certain soil parameters gradually change with deglaciation age: reduced pH values and calcium content, and increased organic matter content [10,45,47]. Another aspect is that several abiotic parameters may be patchily distributed within the same age zone. Local factors like topography, exposition, substrate type, and moisture can modify the succession pattern from the very start [8,9,10,12,31,34,48,49,56,62]. For instance, an open sand and silt substrate without shelter can be unsuitable for carabid beetles. A young pond created by local topography may produce chironomid midges to serve as food for local predators and boost early succession. Unfortunately, arthropod studies in places with many ponds for comparison are lacking in the Alps, due to bedrock characteristics.

We should also add wind as a driving force. In principle, all organisms present on pioneer ground, including algae and mosses, are favored by wind transport. Such transport is episodic, depending on wind strength and direction. Furthermore, shelter from wind may be important for local survival.

A cold and open habitat near the retreating glacier facilitates the establishment of cold adapted and open-ground species among Collembola, spiders, and Coleoptera. Certain competition-weak plants are known to thrive on pioneer ground until vegetation closes [4,6,7,8]. Such negative interactions are, however, little understood in the arthropod communities. Among plants, competition has been assumed to become the driving force for species turnover in later successional stages [5]. There is a species turnover in arthropod succession, but is it driven by competition, or by other inhibiting interactions? Are the generalist predators also competing in the early stages? Is food really limiting, or can they avoid competition by using different microhabitats? These factors are more difficult to study among mobile arthropods than among immobile plants.

### 5.2. How to Quantify Food Choice?

Many potential prey groups can disperse to the pioneer ground by air, either by wind or active flight [19,20,21]. Various field observations and trapping systems in the Austrian case indicated strongly that flying insects were of allochthonous origin. A better quantification between allochthonous and autochthonous prey would improve our understanding of how pioneer food webs are fueled.

Regarding gut analyses, both DNA and direct observations in the microscope are methods with differing strengths and weaknesses. DNA is an elegant method to identify prey taxa but only records presence/absence and does not distinguish between a gut containing, for instance, few Collembola and one containing many. Quantification of a certain prey is therefore by the percentage of stomachs containing the actual sample. A high percentage of Collembola prey can be achieved even if few Collembola were eaten by each predator. The Norwegian microscopic study of gut contents illustrated that. While the percentage of guts containing Collembola was considerable (34% in *Mitopus morio*, 17% in *Bembidion hastii*, and 13% in *Nebria nivalis)*, the mean number per gut was less than one in all predators. As biomass, Collembola prey was insignificant. All three predators also consumed spiders, but always in very low numbers [57].

### 5.3. Pristine Ground—Both a Sink and a New Ecosystem

Pioneer ground functions more or less as an ecological sink. For instance, aphids arriving without any host plant available, will soon die but can represent valuable prey [17]. Certain arthropods found in fallout or on sticky traps near Hardangervidda glacier in Norway clearly did not belong to the pioneer community and died soon after arrival [20]. There are also several examples of arthropods trapped in pitfall traps on Icelandic nunataks that were non-survivors [63]. Dead arthropods may, however, contribute to soil fertilization and facilitate the establishment of plants [22]. If the transport of arthropods is continuous, for instance by ballooning Linyphiids, species may be wrongly recorded as a “resident” even if individuals die shortly after landing. Truly resident, reproductive species may be difficult to identify. However, the presence of larvae and juveniles (which are less mobile), several gravid females, and newly hatched adults, all indicate a reproducing population.

The new ecosystem starts to function as soon as sufficient food sources are present and the population size as well as the sex-ratio of the resident populations is favorable [29]. Collembola, the super-pioneers, evidently have a great ability to find food on newly exposed ground. Most authors have assumed that early Collembola are detritivores, decomposing blown-in dead organic material [13,59,63]. In the Norwegian study of their gut contents, pioneer Collembola were revealed to be mainly herbivores, grazing on diatom algae in biofilm or on tiny pioneer mosses. Only later in succession did certain species shift to a diet of fungal hyphae, typical for decomposers (Table 9) [11,57]. None of the other studies referred to contradict the possibility that pioneer Collembola can be herbivores. For instance, the two Collembola species *Isotoma viridis* and *Orchesella bifasciata* were placed close to the moss-eating beetle *Simplocaria semistriata* in a study of stable ^15^N isotopes [59]. This result was considered a support for Collembola as plant litter decomposers, but an alternative explanation is herbivory. In Iceland, the ^13^C isotopes of Collembola resembled that of plants and lichens [63]. In the DNA-based documentation of an Austrian pioneer food web with Collembola as an important prey, it was concluded, “A logical next step would now be to investigate the food sources of collembolans in more detail as it is not clear how important allochthonous input (detritus) is for these springtails compared to local production by algae and microbes” [17].

### 5.4. The Pioneer Community Enigma: A Mix of Different Life Forms

Within a certain geographical area, the community structure of arthropods on freshly deglaciated ground is rather predictable, at least on higher taxonomic level. But why do certain specific arthropods colonize pioneer ground, while other taxa present in the surroundings do not? One might expect that the ecological filters would result in pioneer species with strong similarities in their ecology. If so, are they specialists or generalists? If specialist, in what way? The key to understand their common success as pioneers is what we have called, “The pioneer community enigma”.

To be a good disperser and find advantageous shelter are obvious favorable properties. However, to understand why they thrive (reproduce and survive), we must look at the ecology of each species. Can species with quite different ecologies live together in a pioneer community? If so, what is the key factor for each of them? And which parameters do not impact colonization?

As a case study, we have chosen to look at ecological parameters for a number of pioneer arthropods near Hardangerjøkulen glacier in Norway. Here, both micro- and macroarthropods have been studied and gut contents analyzed. Furthermore, early chlorophyll was detected.

In Table 10, species from this locality are listed with the earliest colonizers on top, followed by an approximate succession of species downwards. Algae and mosses are included in the list of colonizers. For each organism, we ask whether it is a specialist or a generalist concerning three parameters: climate, habitat, and food. This presentation is rather schematic, but still tells us something about the species’ suitability as pioneers. Two very early species are cold-adapted and probably prefer open space: the Collembola *Agrenia bidenticulata* and the Carabidae *Nebria nivalis.* Both are specialists in climate and habitat, and *Agrenia* also as a biofilm-eater. Their key is their specializations, and they follow the retreating ice edge closely. However, their specializations have a cost: they belong to the near-glacier fauna community, which in the long term is threatened by glacier retreat. Cold-adapted carabid species and other arthropods is also a characteristic feature of pioneer communities in Austrian and Italian Alps. Several carabid species belonging to the genera *Nebria* and *Oreonebria* depend on glaciers [38,39]. Interestingly, until forty years ago, the species *Nebria germari* was particularly abundant in front of the glacier, on scree slopes and on alpine prairies of the southern Alps. Recent studies demonstrated its extinction on alpine prairies and a consequently population contraction [40,64]. Large-sized populations can be still found only on recently deglaciated terrains and on some debris-covered glaciers [39,40]. Conversely, some cold-adapted steno-endemic species belonging to the genus *Oreonebria* (e.g., *Oreonebria lombarda*) were recorded also far from the glacier front, but only on glacier forelands located in peripheral mountain ranges with high winter precipitation rates [10].

A preference for open ground is a key also for the following species in Table 10: the Carabidae beetle *Bembidion hastii*, the Lycosidae spider *Pardosa trailli*, as well as various pioneer mosses, the Collembola *Bourletiella hortensis*, and probably the Byrrhidae beetle *Simplocaria metallica*. The most sensitive among these is probably *B. hastii* which disappears as soon as vegetation closes in. Furthermore, we have moss-eating specialists among the pioneers: *Bourletiella hortensis* and *Simplocaria metallica*, and two omnivore *Amara* species.

Four generalist predators are listed in Table 10. The tolerance for different prey types is their key. Three of them are specialists on open ground, and one is cold adapted. The super-generalist seems to be the Opiliones *Mitopus morio*, with a general high tolerance for variations in climate, habitat, and prey [26]. Likewise, in the Austrian Rotmoos valley, *M. morio* is widespread, apart from on the youngest site where *M. glacialis* takes over. Without looking at the ecology of each species, the enigma could not be solved.

Certain parameters seem to be less important when considering ability to colonize. Reproduction mode could theoretically be important: One might easily assume that parthenogenetic species would be favored as pioneers, since one individual would be sufficient for reproduction and establishment. Furthermore, species with short life cycle and ability to reproduce rapidly, might thought to be favored. It has been shown, however, that pioneer microarthropods are a mix of parthenogenic and sexual species, as well as species with short and long life-cycle [23,30,45]. Why it is so, remains unclear. Furthermore, among pioneer macroarthropods, we find variation in reproduction mode and life cycle length. In Austria, *Mitopus glacialis* is a parthenogenetic species with an annual life cycle. Wolf spiders are sexual with an annual life cycle. The actual carabids have a sexual reproduction and extended life cycles which include an astonishing longevity of adults covering three reproducing seasons (Kaufmann, unpublished) [40].

To summarize, the ecological filters allow quite different life forms to pass, and various types of specialists and generalists live together in pioneer communities. “The pioneer community enigma” can only be understood through close studies of each species’ demands and tolerances.

### 5.5. Nature Conservation Aspects

Cold adapted pioneer arthropods living adjacent to glaciers are in danger of extinction due to rapid glacier melting [10,28,38,39,50,51,52]. Some may survive for a time, either along cold rivers and brooks or in subterranean, cold microhabitats among stones and gravel. Cold adapted species are both ecologically and physiologically interesting. They have adapted to low temperatures during long time periods. Because the Alps avoided a complete ice covering during the last glaciation, cold adapted species not only survived here, but developed new species, including endemic ones in the Alps. There are, for instance, many cold adapted species in the Alps of the carabid genera *Nebria* and *Oreonebria*. There are also a number of cold adapted spiders and Collembola present (Table 3 and Table 5). Furthermore, certain brachypterous beetles with little dispersal capacity and dependent on stable, often cold, habitats, are clearly threatened when glaciers retreat rapidly [50,65]. Where glaciers reach lower altitude, for instance down to the forest line, cold adapted species may still survive in cool, supraglacial debris [66]. In that situation, the specialization of cold adapted species becomes very evident [24]. Protecting refugia that allowed the survival of such species during Holocene climatic optimum, could help such species during the current global warming [24,28].

Our view conforms well with a global meta-analysis of biodiversity change across three major glacier-influenced systems: glacier-fed fjords, glacier-fed freshwaters, and glacier forefields. The analysis concludes that there are both losers and winners following glacier retreat. Most of the losers are specialist species, uniquely adapted to glacial conditions, while winners are typically generalists, and often invasive species [67].

## 6. Conclusions

### 6.1. General Findings

Arthropods colonize deglaciated ground in a rapid succession and may develop food webs within a few years, before higher plants establish or only occur sporadically.Pioneer species are fewer, and succession is slower at high altitudes and latitudes.Each pioneer species has to overcome certain obstacles, or “ecological filters”: It must travel, it must find an acceptable habitat with tolerable climate and shelter to avoid hostile conditions, it must be able to find food, and it must be able to reproduce.Microarthropods, especially Collembola, are “super-pioneers”. They colonize during the very first years, even in high alpine or high arctic conditions.Macroarthropod predators follow closely and are well represented after just a few years (1–5 years). Typical taxa in Europe are Carabidae among Coleoptera, Lycosidae, and Linyphiidae among Aranea, and *Mitopus* sp. among Opiliones.On species level, pioneer arthropods differ between northern Europe and the Alps, due to different glaciation and speciation history. However, certain genera are common to both.A pioneer arthropod community is a mixture of specialists and generalists.The pioneer community may include both parthenogenetic and sexually reproducing species, species with short or long life cycle, and small as well as large species.Certain specialists may disappear after a few decades (30–50 years). Examples are cold-adapted species and species preferring open ground.Melting of glaciers threaten several cold-adapted species within various groups as Carabidae, Araneae, and Collembola.Recent studies challenge the classic “predator first paradox”, which describe a heterotrophic pioneer community depending on airborne prey for predators and airborne dead organic material for decomposing Collembola. It has been shown that predators can be fed by locally produced food, and Collembola can be herbivores.Certain “invisible” carbon sources can contribute to the pioneer community: terrestrial biofilm with diatom algae or cyanobacteria, tiny pioneer mosses, pollen, and bioavailable ancient carbon released by the glacier.Two well-studied pioneer food webs, one from Norway and one from Austria, revealed similarities in arthropod families and genera, but the structure and function of the food webs differed.The pioneer ground can be surprisingly rich in microhabitats and food sources.For many arthropods, pioneer ground is a sink, but dead animals can contribute as prey or soil fertilizers.Definition of succession is a matter of scale. For instance, a small pioneer moss turf may harbor moss-eating Byrrhidae beetles. Moreover, patches with different microclimate or substrate may have different succession pathways.Terrestrial and aquatic environments can be connected in early succession. For instance, young ponds and glacial stream rivers may produce chironomid midges that serve as prey for terrestrial predators.

### 6.2. The Birth of an Ecosystem: Short Answers to the Two Main Questions

A.How “flexible” are arthropod communities in the early phase of primary succession?

We find from our comparative studies, that quite different pioneer arthropod communities are possible (Table 2, Table 3, Table 4 and Table 5) reflecting that primary succession can be flexible and does not have a fixed driving force predicting the outcomes. Species with very different life strategies, including both specialists and generalists, may pass through the “ecological filters” and co-occur in a pioneer community (Table 10). The community structure will depend on available candidate species in surrounding source habitats, and their ability to disperse and establish. Altitude, latitude, climate, and local conditions are important parameters. Prey can be transported by air but may also be produced in situ (for instance Collembola and chironomid midges). Pioneer food webs may rely on different resources, for instance different prey items, early chlorophyll in biofilm or tiny pioneer mosses, or use of bioavailable ancient carbon released by the glacier.

B.Do pioneer arthropod communities under different conditions still have certain basic traits in common?

A common trait for pioneer communities throughout Europe, including high altitude and high Arctic forelands, is the role of Collembola as “super-pioneers”. The “predator first paradox” could be substituted by “The Collembola first principle”. At high altitudes and latitudes, ballooning Linyphiidae spiders represent the macroarthropods. In less extreme climate, pioneer communities of Europe are typically added a macroarthropod association of carabid beetles, Opiliones and Lycosidae, and Linyphiidae spiders. The pristine ground is a sink habitat for many arriving species. However, non-surviving arthropods may serve as prey and also fertilize virgin soils.

## 7. Suggestions for Further Studies

In order to understand “ecosystem birth”, a higher number of young glacier forelands must be investigated. We must learn more about the ecology of pioneer species: Why can certain species act as pioneers, while others, perhaps closely related, cannot? This question is especially relevant for Collembola. Even under harsh conditions, as in high alpine areas and in the high Arctic, certain Collembola species are able to conquer pristine ground immediately. They are able to find shelter and food, and they reproduce. Moreover, they function as prey for early predators, contributing to the first food webs. The good dispersal ability of Collembola combined with a high species diversity is obviously a part of their success story. However, the ultimate question about these super-pioneers, is what do they eat? Do they graze on early chlorophyll rather than being fungal-feeding decomposers, or can they continuously adjust their food according to what is available? Are pioneer Collembola a mix of specialist feeders and/or generalist feeders?

Overlooked early chlorophyll should be searched for, as well as a possible release of bioavailable ancient carbon. It is, however, not only carbon that matters. Nitrogen is important and is commonly considered to be limiting in young ecosystems. It has been claimed that phosphorus may also be a limiting element [68]. On the glacier surface, there is limited primary production (algae and cyanobacteria) and decomposition of organic fall-out (cryoconite holes). This can wash down into the deglaciated ground and favor the upstart of a new ecosystem.

More detailed case studies are needed in order to understand the variation in which an ecosystem is established close to a melting glacier. Comparing cases with similar climate and fauna would illustrate how local conditions may affect food webs; for instance, the presence or absence of ponds. Cases with different climate and fauna from other regions in the world could reveal whether there are some quite general patterns, for instance whether Collembola are general super-pioneers.

Comparison between studies is often problematic due to varying taxonomic resolution. Species-rich but “difficult” groups like Staphylinidae beetles, Collembola, or Acari may not be identified to species level. Furthermore, sample size, type, and number of traps, as well as trapping period, can be critical factors when species number shall be compared between sites. Carabids, for instance, may have their main activity period early in the snow-free season, and trap numbers should be large enough to stabilize the cumulative species number. During soil sampling, local variation should be covered through many small samples rather than a few large ones. Certain large and active, surface-living Collembola easily escape sampling with a soil corer, so pitfall catches represent a valuable supplement.

In order to understand energy flows in pioneer communities, more quantitative data are needed. How large is the contribution of aerial transport of prey compared to local prey production, and how do predators select between the two sources?

Certain neglected invertebrate groups may shortly be mentioned here: Testate amoebae (Protista) are known as early colonizers of high latitudes [14,69]. In four-year-old soil at Hardangerjøkulen glacier in Norway, both Nematoda, Tardigrada, and Rotifera have been documented (Magnusson, Willassen and Hågvar, unpublished). Nematoda and Rotifera have been studied on supraglacial debris of two Alpine debris-covered glacier [70].

Future studies on early animal and plant succession should follow up the geo-ecological perspective, which has been well exemplified in a 33-year chronosequence on the Storbreen glacier foreland, Jotunheimen, southern Norway. It was documented that physical environmental changes, soil development and spatial heterogeneity markedly influenced animal colonization and successional trajectories [71].

Our review represents a mix of conclusions and questions, in a field of increasing attention. The present global shrinking of glaciers gives unique possibilities to reveal both variations, and basic principles, when new ecosystems evolve during primary succession. Such knowledge could shed new light on our ecological understanding.

## Figures and Tables

**Figure 1 insects-11-00644-f001:**
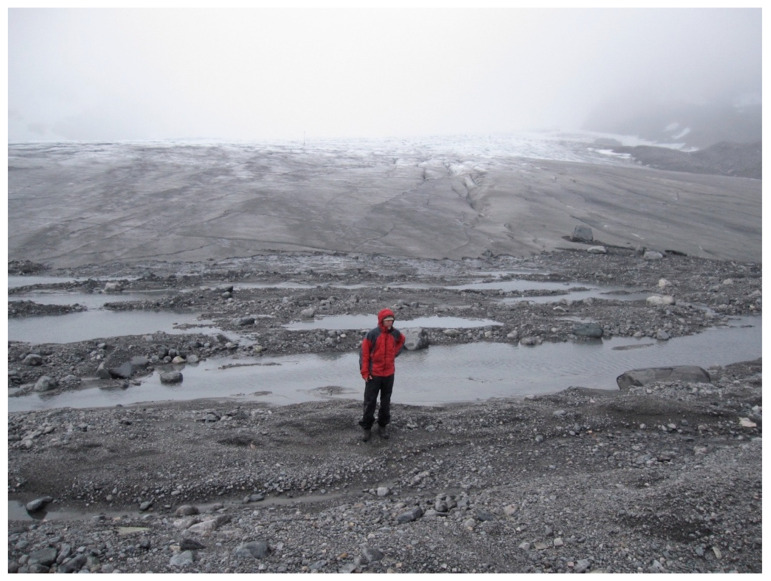
Rapid glacier retreat: In 2010, the Hardangerjøkulen glacier in Southern Norway receded 34 m, corresponding to the area behind the person. From [11]. Photo: Daniel Flø.

**Figure 2 insects-11-00644-f002:**
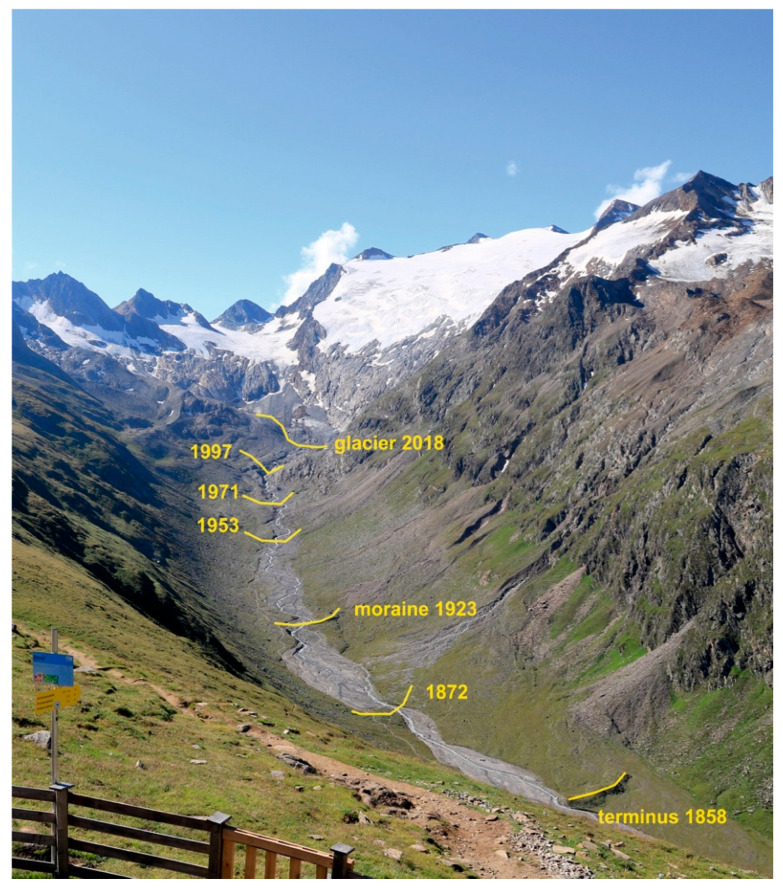
A well-studied site in Ötztal, Austria: The Rotmoos foreland. Past positions of the glacier front are shown. Photo: Rüdiger Kaufmann.

**Figure 3 insects-11-00644-f003:**
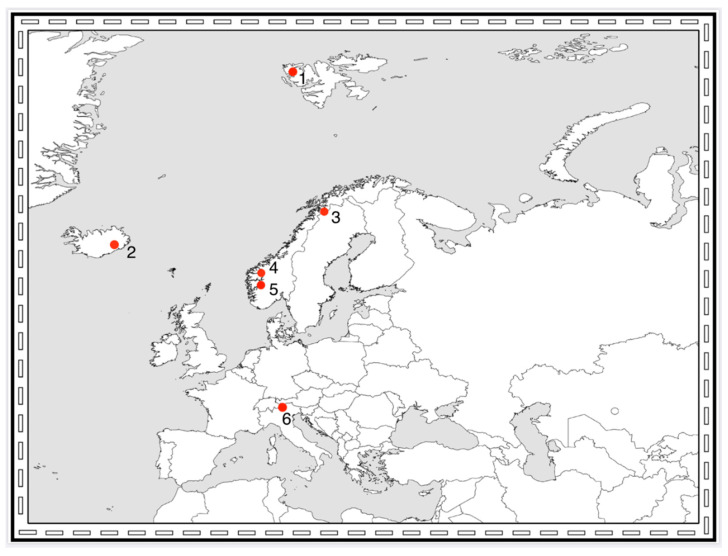
European glacial areas mentioned in this review. 1: Midtre Lovén-bre, Svalbard. 2: Vatnajökull, Iceland. 3: Ålmajallojekna glacier, Sweden. 4: Jostedalen and Jotunheimen, Norway. 5: Hardangerjøkulen, Norway. 6: Closely situated glaciers in the Alps (Italy and Austria).

**Figure 4 insects-11-00644-f004:**
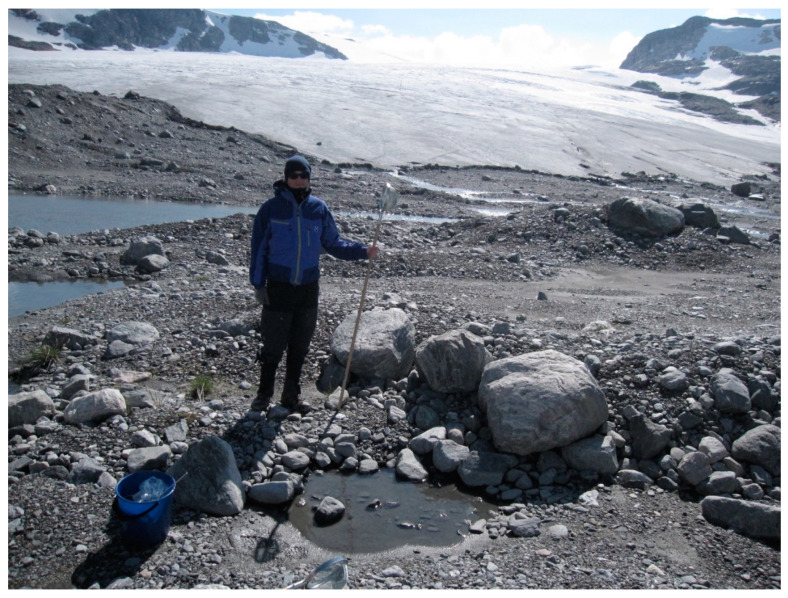
This picture from the foreland of Hardangerjøkulen glacier in southern Norway illustrates that pioneer ground may contain a variety of habitats. From [27].

**Figure 5 insects-11-00644-f005:**
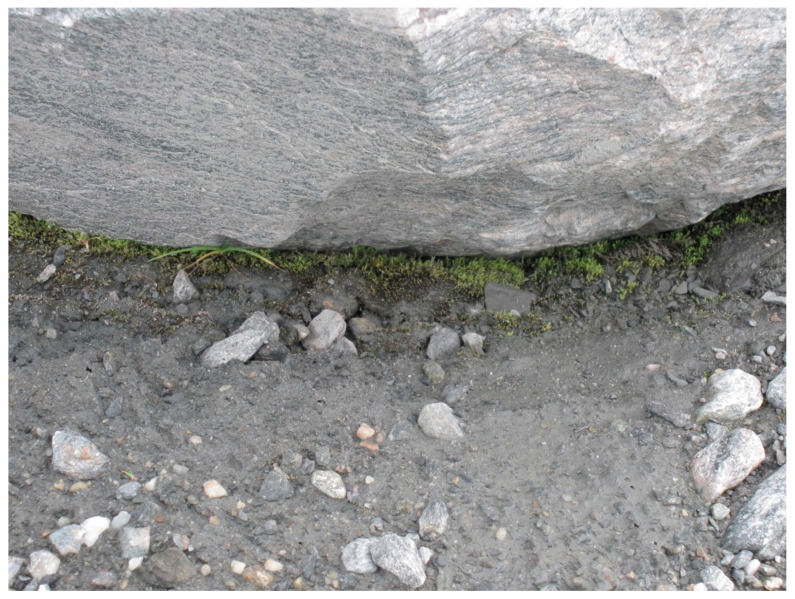
Windblown moss fragments have aggregated along a stone and established a pioneer community before higher plants arrive. From [30].

**Figure 6 insects-11-00644-f006:**
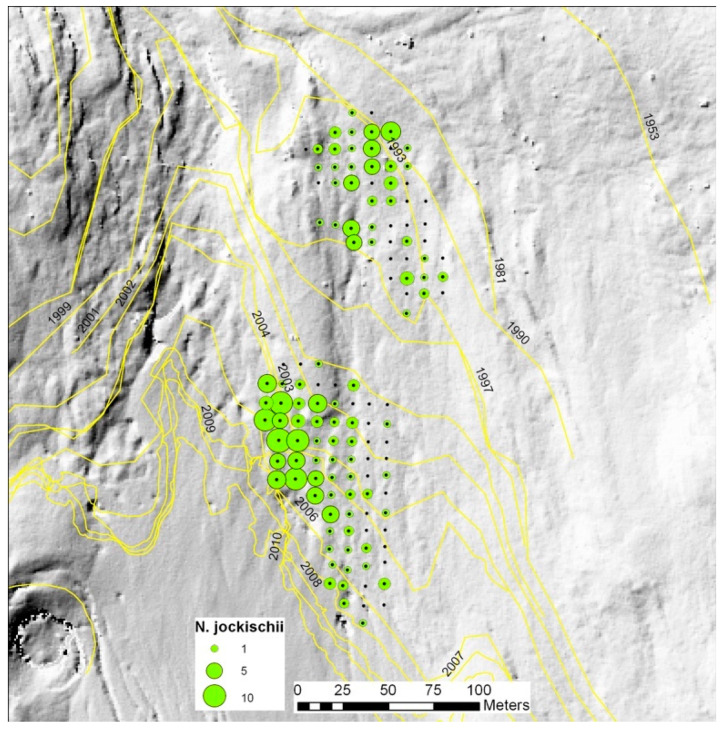
Occurrence of the carabid beetle *Nebria jockischii* in the Rotmoos foreland. Position of the glacier front in different years is indicated. Local conditions are decisive, not only site age.

**Figure 7 insects-11-00644-f007:**
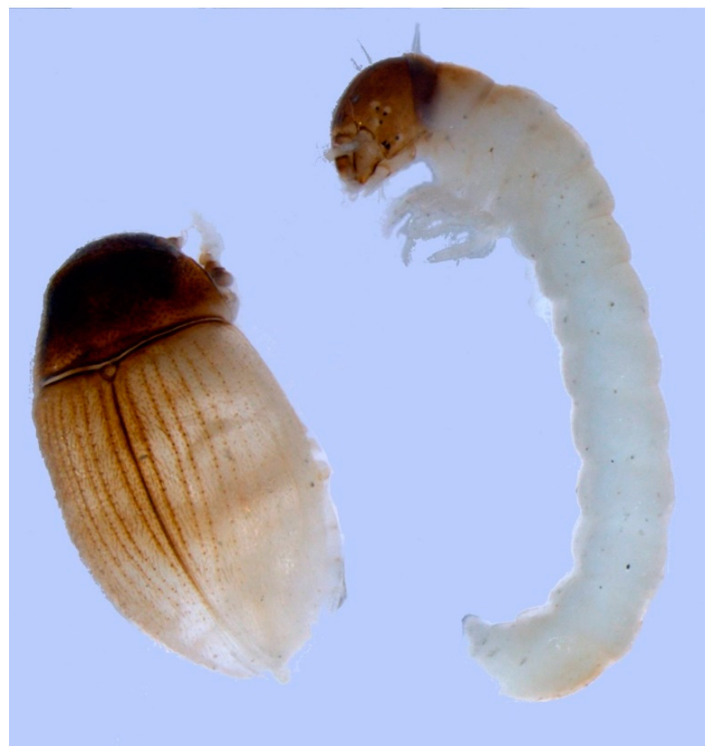
Larva and newly hatched adult of the moss-eating beetle *Simplocaria metallica* (Byrrhidae), extracted from a pioneer moss turf. From [30].

**Figure 8 insects-11-00644-f008:**
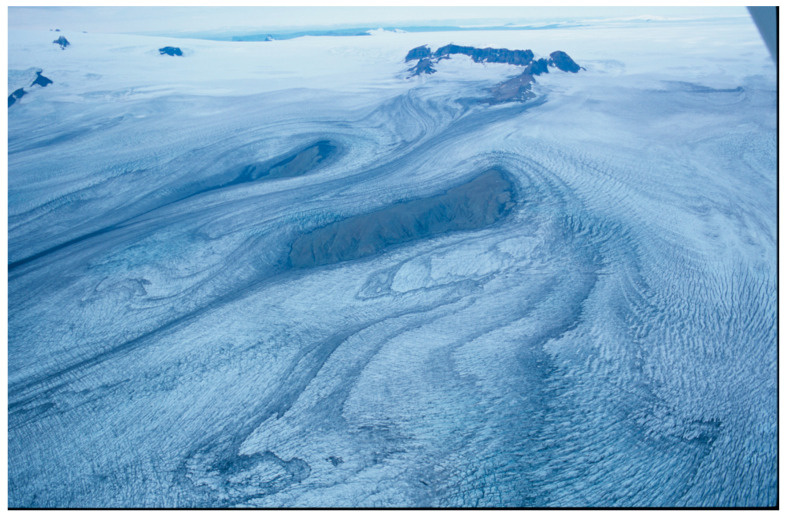
Nunataks on the Vatnajökull glacier, Iceland. The two nunataks in the central part of the photo are Brædrasker (**left**) and Kárasker (**right**). Photo: Oddur Sigurðsson.

**Figure 9 insects-11-00644-f009:**
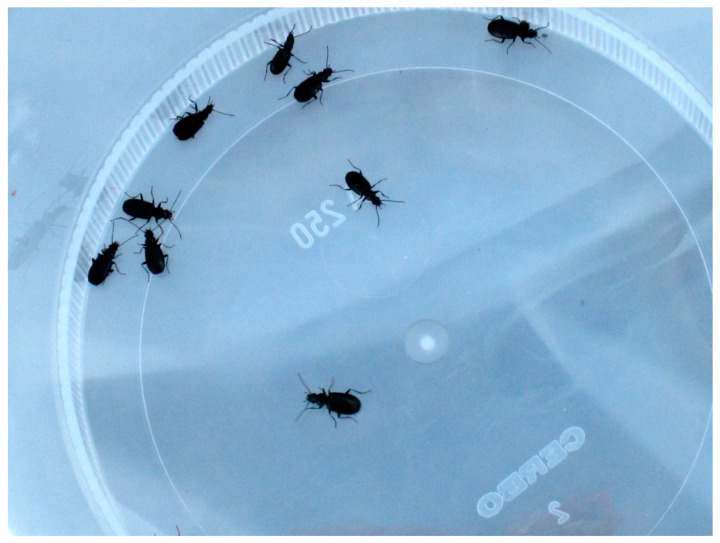
Pitfall traps are efficient in collecting surface active arthropods. Here, the night-active carabid beetle, *Bembidion hastii*, has been trapped alive. From [30]. Photo: Sigmund Hågvar.

**Figure 10 insects-11-00644-f010:**
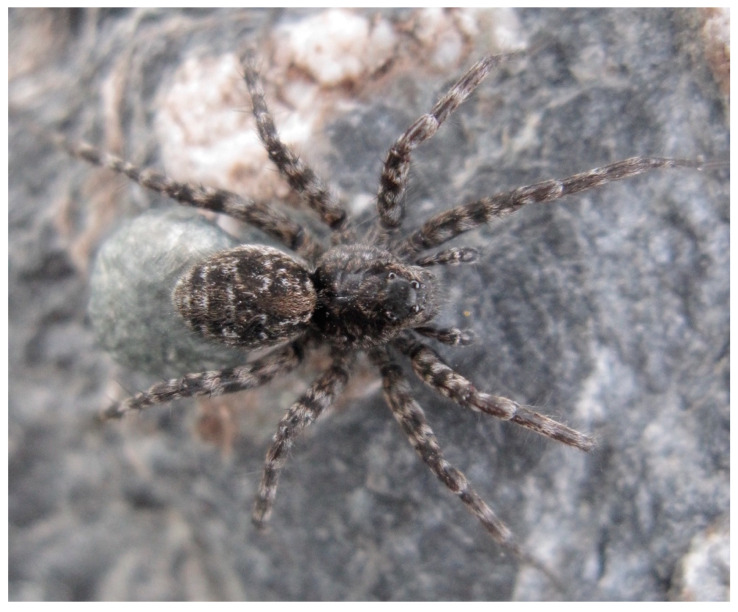
Wolf spiders (Lycosidae) belong to the typical pioneer predator fauna all over Europe, except for arctic and high alpine areas. Here, *Pardosa trailli* from Norway. The animal has a good camouflage and is difficult to discover when standing quietly. Photo: Sigmund Hågvar.

**Figure 11 insects-11-00644-f011:**
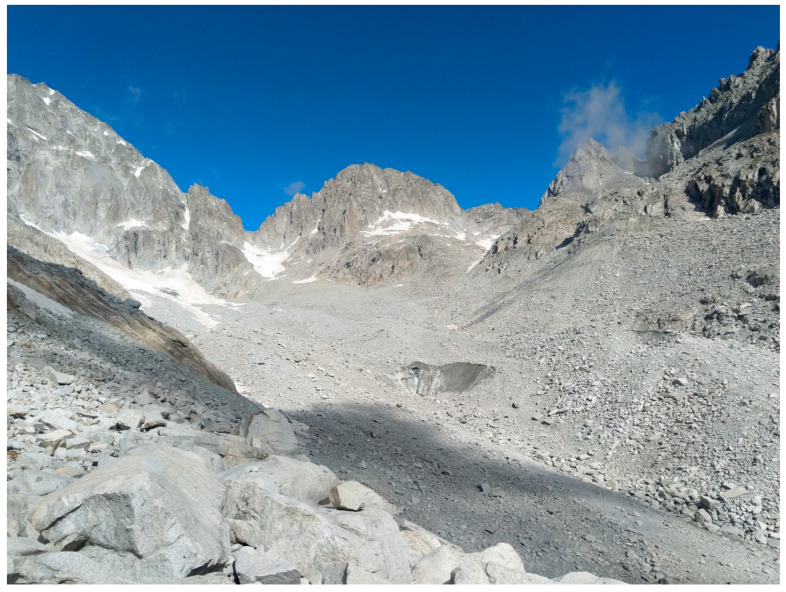
The debris-covered glacier Vedretta d’Amola (Presanella Mountain Group, Italy) and its glacier foreland. Photo: Mauro Gobbi.

**Figure 12 insects-11-00644-f012:**
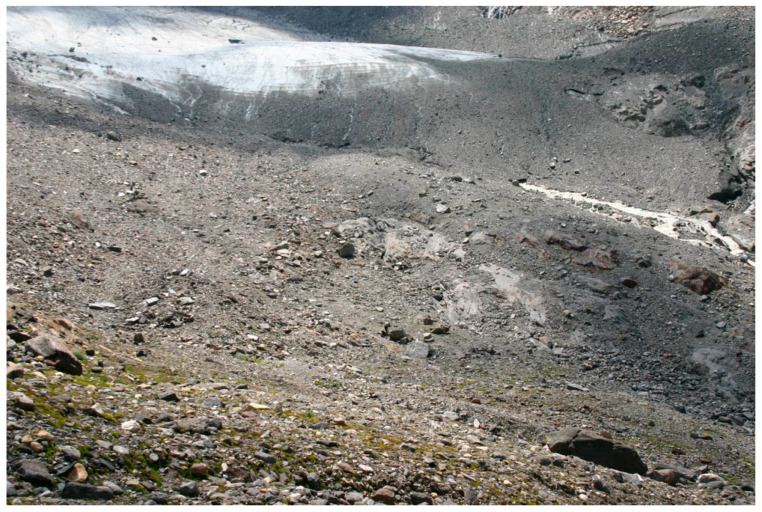
Pioneer ground in the Rotmoos foreland, Austria, where food web studies were performed. Photo: Rüdiger Kaufmann.

**Table 1 insects-11-00644-t001:** Food sources of surface-active arthropods on 3–6 year old ground, based on gut content analyses. Data from Hardangerjøkulen glacier, Norway [11].

Species	Group	Bio-Film	Fungal Hyphae	Bryo-Phytes	Vascular Plants	Arthro-Pods	Ancient Carbon via Chironomidae
*Agrenia bidenticulata*	Collembola	x					
*Desoria olivacea*	Collembola	x					
*Isotoma viridis*	Collembola	x	x				
*Lepidocyrtus lignorum*	Collembola		x				
*Bourletiella hortensis*	Collembola		x	x			
*Simplocaria metallica*	Coleoptera			x			
*Amara alpina*	Coleoptera			x	x	x	
*Amara quenseli*	Coleoptera			x		x	
*Nebria nivalis*	Coleoptera					x	x
*Bembidion hastii*	Coleoptera					x	x
*Mitopus morio*	Opiliones					x	x

**Table 2 insects-11-00644-t002:** Microarthropods recorded in young glacier forelands in Iceland, southern Norway, and Svalbard.

Country and Locality	Iceland, Nunataks	Southern Norway, Hardangerjøkulen			Svalbard		
Altitude (m)	460–728	1400			50		
Reference	[19]	[23,30,45]			[14]		[46]
Age of ground (years)	<10	0	3	32–34	2	16	0
COLLEMBOLA							
SYMPHYPLEONA							
*Bourletiella hortensis*			**X**	**X**			
*Heterosminthurus claviger*	**X**						
*Sminthurides inequalis*					**X**	**X**	
*Sminthurides malmgreni*	**X**						
ENTOMOBRYOMORPHA							
*Agrenia bidenticulata*		**X**	**X**	**X**			**X**
*Desoria infuscata*		**X**	**X**	**X**			
*Desoria olivacea*	**X**		**X**	**X**			
*Desoria tolya*	**X**		**X**	**X**			
*Folsomia brevicauda*	**X**						
*Folsomia quadrioculata*	**X**				**X**	**X**	
*Isotoma anglicana*	**X**				**X**	**X**	
*Isotoma viridis*			**X**	**X**			
*Lepidocyrtus lignorum*	**X**		**X**	**X**			
*Pseudisotoma sensibilis*	**X**						
*Tetracanthella arctica*	**X**						
*Tetracanthella wahlgreni*				**X**			
PODUROMORPHA							
*Ceratophysella* sp.	**X**						
*Ceratophysella scotica*			**X**	**X**			
*Hypogastrura tullbergi*					**X**	**X**	
*Hypogastrura concolor*							**X**
*Mesaphorura* sp.	**X**						
*Micranurida pygmaea*	**X**						
*Oligophorura groenlandica*	**X**						
*Oligophorura schoetti*				**X**			
*Protaphorura* sp.	**X**						
ACARI							
ORIBATIDA							
*Camisia anomia*					**X**	**X**	
*Liochthonius* cf. *sellnicki*				**X**			
*Liochthonius clavatus*	**X**						
*Liochthonius strenzkei*	**X**						
*Pantelozetes paolii*	**X**						
*Tectocepheus velatus*	**X**			**X**		**X**	
PROSTIGMATA	**X**			**X**			
GAMASIDA				**X**	**X**	**X**	

**Table 3 insects-11-00644-t003:** Collembola recorded in young glacier forelands in the Alps.

Country and Locality	W-Sorapiss, Italy			C-Sorapiss, Italy		Agola Glacier, Italy			Amola Glacier, Italy		Rotmoos Glacier, Austria		Hintereis Glacier, Austria	
Altitude (m)	2300–2400			2200–2300		2500–2600			2500–2680		2450		2400	
Reference	Valle et al. (unpublished data)					Valle et al. (unpublished data)			Valle et al. (unpublished data)		[47]	[25]	[42]	
Age of ground (years)	**On ice**	**<50**	**50**	**On ice**	**40**	**On ice**	**<25**	**<50**	**On ice**	**<18**	**<40**	**0–3**	**0–1**	**2–4**
SYMPHYPLEONA														
Symphypleona indet.	**X**	**X**	**X**			**X**	**X**	**X**						
*Heterosminthurus diffusus*	**X**					**X**				**X**		**X**		
*Sminthurinus trinotatus*	**X**													
*Bourletiella repanda*													**X**	**X**
ENTOMOBRYO-MORPHA														
*Orchesella* cf. *alticola*	**X**	**X**	**X**	**X**	**X**	**X**	**X**	**X**		**X**			**X**	**X**
*Orchesella bifasciata*											**X**			**X**
*Orchesella* sp.												**X**		
*Entomobrya* *nivalis*														**X**
Isotomidae indet.	**X**	**X**		**X**	**X**		**X**	**X**						
*Folsomia manolachei*											**X**			
*Isotoma viridis*											**X**			
*Desoria saltans*													**X**	
*Parisotoma notabilis*											**X**			
*Proisotoma schoetti*													**X**	**X**
*Proisotoma crassicauda*													**X**	**X**
*Tetracanthella specifica*											**X**			
*Desoria nivalis*										**X**				
*Pachyotoma crassicauda*							**X**			**X**				
*Pachyotoma pseudorecta*												**X**		
*Isotomurus pseudopalustris*								**X**						
*Isotomurus maculatus*							**X**							
*Isotomurus palliceps*										**X**		**X**		
*Isotomurus alticolus*												**X**		
*Lepidocyrtus curvicollis gr.*												**X**		
*Lepidocyrtus* sp.	**X**		**X**				**X**	**X**	**X**					
*Tomocerus* cf. *minor*					**X**		**X**	**X**						
PODUROMORPHA														
*Pseudachorudina alpina*			**X**					**X**						
*Ceratophysella tuberculata*				**X**										
*Micranurida pygmaea*											**X**			
*Mesaphorura critica*											**X**			
*Hypogastrura* cf. *socialis*										**X**				
*Hypogastrura parva*											**X**			
*Hypogastruridae indet*.												**X**		

**Table 4 insects-11-00644-t004:** Macroarthropods sampled from young glacier forelands of Norway, Sweden, Iceland, and Svalbard. The data are restricted to ground-dwelling beetles (Coleoptera), harvestmen (Opiliones), and spiders (Araneae). Only few of the Staphylinidae from Jostedalen and Jotunheimen were identified. Flying insects, as Diptera and Hymenoptera, are not included, as they may be casual visitors.

Country and Locality	Jostedalen, Norway				Jotunheimen, Norway				Hardanger-Vidda, Norway		North-Ern Swe-Den	Ice-Land	Sval-Bard	
Glacier name	Aus-ter-dals-breen	Berg-set-breen	Fåberg-støls-breen	Bø-dals-breen	Stygge-dals-breen	Bøver-breen	Stor-breen	Vesle-juv-breen	Hardanger-jøkulen		Ålma-jallo-jeknaglacier	Nu-na-taks	MidtreLovén-bre	
Altitude (m)	320–390	400–500	480–620	560–600	1270	1400	1400	1860	1400		1180–1344	460–728	50	
Reference [numbers]	[48]	[48]	[48]	[48]	[48]	[48]	[48]	[48]	[36]		[49]	[19]	[14]	
Climatic zone	Sub-alpine	Sub-alpine	Sub-alpine	Sub-alpine	Low alpine	Low/mid-alpine	Low/mid-alpine	High alpine	Mid-alpine		Low/mid-alpine	Arctic	High arctic	
Age of ground (years)	<20	<20	<20	<20	<20	<20	<20	<20	3	40	<40	<10	2	16
COLEOPTERA, CARABIDAE														
*Amara alpina*		x	x	x		x	x		x	x				
*Amara quenseli*			x				x		x	x		x		
*Nebria* sp.	sp.		sp.	sp.	*N. nivalis*	sp.	sp.		*N. nivalis*	*N. nivalis* *N. rufescens*	*N. nivalis* *N. rufescens*			
*Bembidion* sp.			*B. fellmanni*		*B. fellmanni*		*B. fellmanni*		*B. hastii*	*B. hastii*	*B. hastii*			
*Patrobus septentrionis*										x				
*Notiophilus aquaticus*					x	x								
*Miscodera arctica*					x		x	x						
COLEOPTERA, BYRRHIDAE														
*Simplocaria metallica*									x	x				
*Byrrhus arietinus*						x	x							
COLEOPTERA, CURCULIONIDAE														
*Otiorhynchus nodosus*										x				
*Otiorhynchus arcticus*												x		
COLEOPTERA, STAPHYLINIDAE														
*Oxypoda annularis*										x				
*Boreaphilus henningianus*										x				
*Geodromicus longipes*									x	x				
*Acidota crenata*			x											
*Anthophagus alpinus*											x			
*Arpedium quadrum*									x		x			
*Coryphiomorphus hyperboreus*											x			
*Olophrum boreale*											x			
*Tachinus elongatus*											x			
*Quedius* sp.			x					x						
COLEOPTERA, HYDROPHILIDAE														
*Helophorus glacialis*										x				
COLEOPTERA, DYTISCIDAE														
*Agabus bipustulatus*									x					
*Agabus thomsoni/lapponicus*											x			
COLEOPTERA, CHRYSOMELIDAE														
*Phratora polaris*											x			
COLEOPTERA, NITIDULIDAE														
*Meligethes aeneus*											x			
COLEOPTERA, SILPHIDAE														
*Thanatophilus lapponicus*											x			
COLEOPTERA, LEIODIDAE														
*Catops tristis*			x				x							
OPILIONES, PHALANGIIDAE														
*Mitopus morio*	x	x	x	x	x	x	x		x	x	x	x		
ARANEAE, LYCOSIDAE														
*Pardosa trailli*						x	x		x	x				
*Pardosa palustris*											x	x		
*Pardosa pullata*			x											
ARANEAE, GNAPHOSIDAE														
*Zelotes subterraneus*		x												
*Gnaphosa leporina*											x			
*Micaria alpina*											x			
ARANEAE, LINYPHIIDAE														
*Gonatium rubellum*								x						
*Oedothorax retusus*											x			
*Erigone longipalpis*			x			x	x	x						
*Erigone tirolensis*									x			x		
*Erigone psychrophila*													x	x
*Erigone arctica*									x		x			x
*Meioneta nigripes*											x	x	x	x
*Collinsia spetsbergensis*												x	x	x
*Collinsia holmgreni*									x		x	x		
*Islandiana princeps*												x		
*Improphantes complicatus*												x		

**Table 5 insects-11-00644-t005:** Macroarthropods sampled from young glacier forelands of the Alps (Italy and Austria). The data are restricted to ground-dwelling beetles (Coleoptera), harvestmen (Opiliones), and spiders (Araneae). Flying insects, as Diptera and Hymenoptera, are not included, as they may be casual visitors. All data are from the Alpine zone, above the tree line.

Country and Glacier Name	Rotmoos-tal, Austria	Cedec Glacier, Italy	Forni Valley, Italy	Forni Valley, Italy		Trobio Glacier, Italy	W-Sorapiss Glacier, Italy			C-Sorapiss Glacier, Italy		D’Agola Glacier, Italy				Amola Glacier, Italy		Hintereis, Austria		Hornkees, Austria	
Altitude (m)	2500	2700	2500	2500		2500	2300–2400			2200–2300		2500–2600				2500–2680		2400		2200	
Reference	[31] and Kauf-mann, unpubl.	[50]	[38]	[51,52,53]		[10]	[43]			[43]		Gobbi et al. unpubl.				[39]		[42]		[41]	
Age of ground (years)	<10	<20	<5	On ice	24	<30	On ice	<50	50	On ice	40	On ice	<20	<50	50	On ice	<18	0–1	2–4	2	6–10
COLEOPTERA, CARABIDAE																					
*Nebria germari*	X						X	X	X	X	X	X	X	X	X	X	X			X	X
*Nebria jockischii*	X		X														X	X		X	X
*Nebria hellwigi*																					X
*Nebria rufescens*	X																				X
*Oreonebria castanea*	X	X	X	X																	
*Oreonebria diaphana*							X	X	X	X	X										
*Oreonebria soror*						X											X				
*Amara quenseli*	X	X			X													X			
*Amara erratica*																					X
*Pterostichus jurinei*	X																				
*Carabus sylvestris*		X			X																
*Carabus adamellicola*													X	X	X						
*Trechus dolomitanus*									X		X										
*Sinechostichus doderoi*			X																		
*Princidium bipunctatum*					X																
*Bembidion geniculatum*																		X			
COLEOPTERA, BYRRHIDAE																					
*Simplocaria semistriata*	X																				
COLEOPTERA, ELATERIDAE																					
*Fleutiauxellus maritimus*	X			X														X	X		
COLEOPTERA, HYDROPHILIDAE																					
*Helophorus glacialis*	X																				
COLEOPTERA, STAPHYLINIDAE																					
*Eusphalerum anale*	X																				
OPILIONES, PHALANGIIDAE																					
*Dicranopalpus gasteinensis*	X																				
*Mitopus glacialis*	X																	X			
*Gyas annulatus*																		X			
ARANEAE, LYCOSIDAE																					
*Pardosa nigra*	X															X	X	X	X		
*Acantholycosa pedestris*							X	X	X	X	X		X								
*Pardosa saturatior* ARANEAE,	X			X																	
GNAPHOSIDAE																					
*Drassodex heeri*									X					X	X						
ARANEAE, ERIGONIDAE																					
*Erigone atra*													X								
*Erigone tirolensis*	X																				
*Mecynargus paetulus*	X																				
*Entelecara media*	X					X															
*Walckenaeria vigilax*	X																				
ARANEAE, THERIDIIDAE																					
*Robertus arundineti*	X																				
ARANEAE, AGELENIDAE																					
*Thanatus formicinus*					X																
*Coelotes mediocris*					X																
*Coelotes pickardii* *tirolensis*														X	X						
ARANEAE, THOMISIDAE																					
*Xysticus audax*					X																
*Xysticus alpinus*									X	X	X										
ARANEAE, LINYPHIIDAE																					
*Meioneta rurestris*	X																				
*Meioneta gulosa*	X																				
*Meioneta nigripes*	X																				
*Lepthyphantes variabilis*	X																				
*Agyneta rurestris*						X	X	X					X			X					
*Mughiphantes brunnerii*							X			X											
*Mughiphantes handschini*													X	X	X						
*Mughiphantes* cf. *merretti*														X	X						
*Mughiphantes pulcher*						X															
*Tenuiphanthes jacksonoides*							X														
*Diplocephalus helleri*													X	X	X		X	X	X		
*Troglohyphantes fagei*									X		X										
*Erigone dentipalpis*																	X				
*Oreonetides glacialis*																	X				
*Janetschekia lesserti*																			X		

**Table 6 insects-11-00644-t006:** Main food items for pioneer predators in the Norwegian and the Austrian study, based on gut contents.

Country and Reference	Method	Collembola	Chironomid Midges	Other Predators	Foreign Prey?	Locally Produced Prey?
Norway [57]	Microscope study of gut content	Rare	Common	Rare	Yes, but rare?	Yes, locally produced chironomid midges
Austria [17]	DNA study of gut content	Common	Rare	Common	Yes, Mostly Dipterans and aphids	Yes, locally produced Collembola, and maybe other predators

**Table 7 insects-11-00644-t007:** Quantitative data on Collembola in Rotmoos glacier foreland at three sites with different age [25]. Totals from all sampling methods.

Family	Genus/Species	0–3 Years	9–14 Years	18–25 Years	On Glacier	Total Numbers
Bourletiellidae	*Heterosminthurus diffusus*	1927	320	32	5	2284
Entomobryidae	*Orchesella* sp.	375	61	9	2	447
	*Lepidocyrtus* cf. *curvicollis*	10	104	114	2	230
Isotomidae	*Pachyotoma pseudorecta*	33	1	3	0	37
	*Isotomurus* cf. *alticolus*	57	1	41	1	100
	*Isotomurus palliceps*	1	24	11	0	36
	*Agrenia bidenticulata*	0	0	1	1	2
	*Desoria* sp.	0	0	0	2	2
	Isotomidae indet.	0	0	9	0	9
Hypogastruridae	Hypogastruridae sp. 1	6	4	0	0	10
	Hypogastruridae sp. 2	0	0	0	3	3
	Hypogastruridae sp. 3	0	0	1	0	1
	*Hypogastrura* sp.	0	0	0	1	1

**Table 8 insects-11-00644-t008:** Collembola succession on barren ground close to a Norwegian receding glacier: Percentage dominance of each species at different ages of the ground. From [30].

Age (Year)	0	3	32–36	41–47
Sampling method	Flotation	Pitfall	Pitfall	Pitfall
*Agrenia bidenticulata*	84.6	24.7		
*Desoria infuscata*	15.4	1.5	6.5	
*Bourletiella hortensis*		59.9	1.1	
*Isotoma viridis*		5.2	28.3	21.3
*Lepidocyrtus lignorum*		0.4	50.0	65.3
*Desoria olivacea*		8.0	8.7	4.5
*Desoria tolya*		0.2	4.3	3.0
*Ceratophysella scotica*		0.1	1.1	6.1
Number of animals sampled	26	1465	92	66

**Table 9 insects-11-00644-t009:** Changes in visible gut content during succession in two Collembola species. Relative amounts of different elements are roughly indicated by + (very little) up to ++++ (dominant gut content). Simplified from [11].

Species	Gut Content	3 Years	30–40 Years	63 Years
*Isotoma viridis*	Mineral particles	++++	++++	+
	Diatom algae	+	-	-
	Fungal hyphae	+	++	++++
	Fungal spores	+	++	-
*Lepidocyrtus lignorum*	Mineral particles	++++	++++	+
	Diatom algae	-	-	-
	Fungal hyphae	+	++	+++
	Fungal spores	-	++	+++

**Table 10 insects-11-00644-t010:** This Table illustrates how various specialists and generalists together can shape a pioneer arthropod community. Data from 0 to 7 year old foreland of Hardangerjøkulen glacier, Norway. Illustrations from [11,27].

	Species	Group	Climate	Habitat	Food
TERRESTRIAL	Specialist	Generalist	Specialist	Generalist	Specialist	Generalist
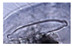	Diatom algaein biofilm	Algae			Open			
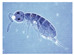	*Agrenia bidenticulata*	Collem-bola	Cold adapted		(Open)		Biofilm	
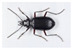	*Nebria nivalis*	Cara-bidae	Cold adapted		(Open)			Generalist predator
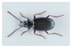	*Bembidion hastii*	Cara-bidae			Open			Generalist predator
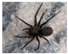	*Pardosa trailli*	Lyco-sidae			Open			Generalist predator
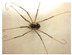	*Mitopus morio*	Opili-ones		Generalist		Generalist		Generalist predator
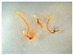	Pioneer mosses	Moss			Open			
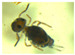	*Bourletiella hortensis*	Collem-bola			Open		Moss	
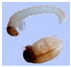	*Simplocaria metallica*	Byrr-hidae			(Open)		Moss	
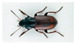	*Amara alpina*	Cara-bidae	(Alpine)				(Moss)	Omnivore
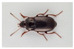	*Amara quenseli*	Cara-bidae					(Moss)	Omnivore
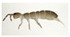	*Isotoma viridis*	Collem-bola					Biofilm and fungal hyphae	
**AQUATIC**
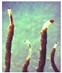	Chironomidae larvae	Diptera			Young ponds		Bio-available ancient carbon	
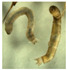	Tipulidae larvae	Diptera			Young ponds		Bio-available ancient carbon	
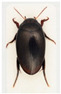	*Agabus bipustulatus*	Dytis-cidae			Young ponds		Predator on Chirono-mid larvae

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
