# Peer review of "Ecosystem Birth near Melting Glaciers: A Review on the Pioneer Role of Ground-Dwelling Arthropods"

_insects, 2020, doi:10.3390/insects11090644_

Round 1
Reviewer 1 Report
I have attached the manuscript with some minor comments, mostly related to grammar and typos.
Overall, this paper represents a unique addition to the literature of an emerging and understudied area of research, and in the process does help shed light on what is know and identifies many additional questions. As such, I feel that it is a significant contribution. The overall style and presentation of the review at times is a bit conversational but is exceptionally readable and does a good job of synthesizing the information overall. I suggest going back through for a thorough look at grammar.

Reviewer 2 Report
This is a review coverning a very interesting topic of insect ecology and ecosystem ecology, moreover quite relevant in these days of large retreat of glaciers. As such it definitely belongs to this topical issues under preparation and is recommended to be published in INSECTS. The highlight of the manuscript in its current form are the General findings in the Conclusions, and the chapter defining the directions for future research.
I am not a specialist in the field, hence my comments will be more from the point of a general entomologist reader.
My biggest problem with this manuscript is that it is something like a review combined with author´s own data. But it is very difficult to trace which parts concern published data, which parts are conclusions based on these data and just adopted from the previous studies, which are conclusions based on previous data made by the current authors, and which are the new data used in this study. I would strongly suggest the authors to polish their review better to make these things more easy to trace. Maybe, a good thing would be to make a table of studies reviewed + new studies presented here a small map showing the locations where they were conducted.
The text of the review is mostly quite easy to follow, but in general seems to be too "wordy" and many information in fact repeat several times through the text. As a final result of that, after finishing the reading, I have many interesting of information in my head, but in a rather unorganized way. I believe that making the text slightly shorter, following more clearly a "one line of story" and writing things slightly more concisely would help a lot.
There are also many large tables through the manuscript, sometimes not really easy to follow (for example, first you have two table of collembolans on the studied sites labelled "microarthropods"), then tables also comparing "microarthropods" but actually only focused on beetles, showing similar information as the collembolan tables, but being formatted differently... What I am totally missing are some summary graphics which would actually present these summary data in some easy-to-understand graphical way, as this is what I would expect from a review much more than large tables repeating the already published data (some of them are important anyway, so I am not suggesting to remove the current table, juts to adapt their formatting and legend and accompany them with some easy to read summary graphics.
There are few more things I commented directly in the manuscript.
For the reasons above, I am suggesting a major revision of the text, in order to make it easier to read and more concise. The topic covered and studies excerpted are OK, this needs no change.
The popular abstract is very poorly written and needs to be considerably improved.
Also, try to be scientific, as this is a scientific paper in a scientific journal. I believe that statements as "Nature´s flexiblity to do something" do not belong to a scientific text. If the comminity is flexible in some aspects, this needs to be specified properly (what is the aspects which is flexible, why, how this is achieved) and not in the romantized way of Mother Nature being flexible in her activities!

Round 2
Reviewer 2 Report
The new version of the manuscript is largely improved which I greatly appreciate, and all my previous comments were addressed properly. I hence have only a few comments on this version:
- I still believe that a map showing the principal localities compared would be very useful. I would just show the localities mentioned in tables 4 and 5, and no problem that localities in the Alps will stand as one point in case the Austrian and Italian sampling sites are very close to each other. The map will also highlight the two geographic areas you compare, which is in fact the center-piece of your study.
- My concern about "nature is flexible" was actually not about the word flexible (that is totally fine), but in using Nature as an active personified subject (and now I noticed that this is not the only case, elsewhere in the text there are phrases like "Nature sorted...", "nature starts up a new environment" or "nature´s vulnerability". In all these cases you take Nature as something like "mother Nature" which actively is doing something, which is I guess the idea and writing style not appropriate to scientific biology. There is nothing like "mother Nature" and all your phrases putting nature in such an active role in fact refer to other subjects. The succession (i.e. the process of forming) of pioneer communities is flexible, not the nature. Arriving species are sorted by environmental conditions, not by a Nature. And so on... In fact, if you take all your sentences in which nature stands as a active subject, remove nature and put the real thing which is active in any of this case (as succession and environmental conditions in examples listed above), your text will be much easier to read, since the active subject will be not an abstract "nature" but something more specific.
- I think the header of chapter 3.3 is little bit confusing in "Northern and Southern Europe". Alps are definitely not what I would call southern Europe, and moreover you refer to that area as "the Alps" on most other cases in the paper. Please keep consistent, to make the paper easy to read. I did not check whether a similar problem is also elsewhere, so please make sure that you stick to "Northern Europe" vs. "the Alps" in all cases.
